# Transcriptome Profiling of Vero E6 Cells during Original Parental or Cell-Attenuated Porcine Epidemic Diarrhea Virus Infection

**DOI:** 10.3390/v15071426

**Published:** 2023-06-23

**Authors:** Ouyang Peng, Yu Wu, Fangyu Hu, Yu Xia, Rui Geng, Yihui Huang, Siying Zeng, Guangli Hu, Chunyi Xue, Hao Zhang, Yongchang Cao

**Affiliations:** State Key Laboratory of Biocontrol, Life Sciences School, Sun Yat-Sen University, Guangzhou 510275, China

**Keywords:** porcine epidemic diarrhea virus, cell-attenuated strain, vaccine, transcriptome

## Abstract

Porcine epidemic diarrhea virus (PEDV) has led to significant economic losses in the global porcine industry since the emergence of variant strains in 2010. The high mutability of coronaviruses endows PEDV with the ability to evade the host immune response, which impairs the effectiveness of vaccines. In our previous study, we generated a highly cell-passaged PEDV strain, CT-P120, which showed promise as a live attenuated vaccine candidate by providing satisfactory protection against variant PEDV infection in piglets. However, the mechanism by which the attenuated CT-P120 adapts to cells during passage, resulting in increased replication efficiency, remains unclear. To address this question, we conducted a comparative transcriptomic analysis of Vero E6 cells infected with either the original parental strain (CT-P10) or the cell-attenuated strain (CT-P120) of PEDV at 6, 12, and 24 h post-infection. Compared to CT-P10, CT-P120 infection resulted in a significant decrease in the number of differentially expressed genes (DEGs) at each time point. Functional enrichment analysis of genes revealed the activation of various innate immune-related pathways by CT-P10, notably attenuated during CT-P120 infection. To validate these results, we selected eight genes (TRAF3, IRF3, IFNL1, ISG15, NFKB1, MAP2K3, IL1A, and CCL2) involved in antiviral processes and confirmed their mRNA expression patterns using RT-qPCR, in line with the transcriptomic data. Subsequent protein-level analysis of selected genes via Western blotting and enzyme-linked immunosorbent assay corroborated these results, reinforcing the robustness of our findings. Collectively, our research elucidates the strategies underpinning PEDV attenuation and immune evasion, providing invaluable insights for the development of effective PEDV vaccines.

## 1. Introduction

Porcine epidemic diarrhea virus (PEDV) is an enveloped, single-stranded, positive RNA virus, belonging to the order *Nidovirales*, family *Coronaviridae,* and genus *Alphacoronavirus* [1]. In general, PEDV could cause porcine epidemic diarrhea (PED) diseases in pigs of all ages, mainly characterized by diarrhea, vomiting, dehydration, anorexia, weight loss, and elevated mortality in piglets, which occurs more regularly in winter and spring [2,3]. Clinical symptoms caused by PEDV infection vary in pigs of all age ranges; however, the mortality rate of PEDV in piglets, especially in neonatal piglets within 10 days, typically reached 100% [4]. In 1978, following the isolation of the first strain, CV777, PEDV was identified as the antigen responsible for the PED outbreak in Belgium [5]. Since then, PEDV has gradually spread and exploded in Western European countries such as Belgium and France, as well as in Southeast Asian countries such as South Korea and Japan [3,6]. Prior to 2010, all epidemic PEDV strains were classified as genotype 1 (G1), such as the classical CV777 and DR13 strains. In October 2010, an extremely pathogenic variant of PEDV belonging to genotype 2 (G2) rapidly swept through the Chinese pig industry, causing the worst epidemic disaster in the country’s pig farming history [2]. Subsequently, the variant strain caused a pandemic in the United States in the spring of 2013 and spread to Canada and Mexico, as well as additional Southeast Asian countries [7,8]. To date, PEDV has been identified as the dominant antigen responsible for nearly all cases of annual diarrheal disease in pigs [9].

Vaccination is considered an effective method to combat PEDV infection. Several live attenuated and inactivated vaccines against classical PEDV strains, such as PEDV CV777, DR13, and KPEDV-9, have been developed and commercially used in numerous countries [4,10]. In contrast, live attenuated vaccines provide a higher protection rate than inactivated or genetically engineered subunit vaccines. The attenuation of PEDV strains on Vero E6 cells for use in commercial vaccines is common across countries. In 1998, Tong et al. passaged PEDV CV777 to 90 generations through the same attenuation pathway and used animal experiments to demonstrate that its virulence was attenuated and remained stable, with excellent immunogenicity, thus preparing China’s first attenuated virus vaccine against swine epidemic diarrhea [11]. Due to antigenic and genetic differences between the vaccine and the pandemic strain, these traditional vaccines gradually lose their better protection and resistance to the mutated and highly pathogenic PEDV strain. Therefore, in our previous study, PEDV CT was subcultured on Vero E6 cells for 120 passages and tested by in vitro testing [12]. The most effective way to prevent PED is through vaccination, in which live attenuated vaccines play a crucial role. A fully attenuated PEDV strain has been obtained by cell passage and may serve as a potential vaccine candidate. Considering that attenuated PEDV can cause more pronounced cytopathic changes and achieve higher virulence than the original parent strain, the CT-P120 strain could serve as a valuable material for studying the pathogenicity and immune evasion mechanisms of PEDV.

Transcriptomic sequencing of eukaryotes cells or tissues is widely used in the fields of gene expression level analysis, differential expression analysis, novel gene mining, single nucleotide polymorphism identification, and gene function annotation [13]. In recent years, high-throughput RNA sequencing (RNA-seq) technology has been applied more and more in the field of viral infection and disease, and has become an effective research tool to reveal the gene interaction network, understand the host–pathogen relationship, and develop new strategies for treatment and prevention interventions [14,15]. Vero E6 cells are particularly well-suited cell models for studying the pathogenesis of viruses [16,17] and are commonly used to produce vaccines [18]. The continual increase of viral titer of PEDV was observed upon passaging in Vero E6 cells [19,20]; however, the mechanism of how PEDV adapts to cells and gains higher proliferation efficiency remains unclear. Furthermore, research on differences in transcriptomic characteristics of parental or cell-attenuated PEDV-infected cells in the early infection stage is lacking.

In this study, we employed RNA sequencing to explore host-pathogen interactions in Vero E6 cells during infection with either the mock, PEDV original parent CT-P10, or cell-attenuated CT-P120 strains. Our data reveal that the attenuated PEDV strain CT-P120 induces fewer differentially expressed genes compared to CT-P10, and the innate immune response of cells is also attenuated, which may contribute to the higher viral load of CT-P120 in Vero E6 cells. Collectively, our study finds potential evidence for immune evasion of coronavirus PEDV and establishes the basis for PEDV vaccine design and therapeutic targets.

## 2. Materials and Methods

### 2.1. Viruses, Cells, and Antibody

The PEDV strains used in this study, CT-P10 and CT-P120, were previously isolated and preserved in our laboratory [12]. Vero E6 cells, sourced from the American Type Culture Collection (ATCC: CCL-81), were routinely cultured in Dulbecco’s Modified Eagle’s Medium (DMEM) supplemented with 10% fetal bovine serum (Invitrogen, Waltham, MA, USA) and 1% antibiotic solution (100 μg/mL penicillin, 100 μg/mL streptomycin, and 25 μg/mL Fungizone^®^; Gibco™, Grand Island, NY, USA). The cells were seeded in culture plates at a density of 5 × 10^6^ cells per plate and maintained for 48 h in a humidified incubator at 37 °C with 5% CO_2_. PEDV challenge was initiated when the cells reached 90% confluence. A mouse monoclonal antibody against PEDV spike protein (S) was prepared and stored in our laboratory.

### 2.2. Indirect Immunofluorescence Assay (IFA)

To determine whether Vero E6 cells were infected with PEDV, the cells were inoculated with a multiplicity of infection (MOI) of 0.1 and incubated for 6, 12, and 24 h post-infection (hpi). At each time point, the cells were washed three times with phosphate-buffered saline (PBS) and fixed with 4% paraformaldehyde at room temperature for 15 min. To enable antibody access to intracellular antigens, the cells were permeabilized with 0.2% Triton X-100 for 15 min and then blocked with 3% bovine serum albumin for 1 h at room temperature. The cells were incubated with a primary antibody, mouse anti-PEDV S monoclonal antibody, and a secondary antibody, an FITC-labeled goat anti-mouse IgG antibody. The cell nuclei were stained with 4′,6-diamidino-2-phenylindole for 5 min at room temperature in the dark. After washing the cells thrice with PBS, they were observed under a fluorescence microscope to assess for infection.

### 2.3. RNA Extraction and Sequencing

Initially, total RNA was extracted from cell samples using Trizol (Takara Bio, Otsu, Japan) according to the manufacturer’s instructions. The concentration and purity of the RNA were subsequently measured using a NanoDrop 2000 spectrophotometer (Thermo Fisher, Waltham, MA, USA). RNA integrity was assessed by agarose gel electrophoresis and the RNA integrity number (RIN) was determined using an Agilent2100 Bioanalyzer. For a single library construction, we ensured that the total RNA amount was ≥1 ug, the concentration was ≥35 ng/μL, OD260/280 was ≥ 1.8, and OD260/230 was ≥1.0.

We then enriched the mRNA from the total RNA using Oligo (dT) magnetic beads, which target the polyA tail structure at the 3′ ends of the mRNA. The enriched mRNA was subsequently fragmented randomly by adding fragmentation buffer, and fragments of approximately 300 bp were selected using magnetic beads.

With the aid of reverse transcriptase (Takara Bio, Otsu, Japan) and random hexamer primers, the first-strand cDNA was synthesized using mRNA as the template. After removing the RNA template, the second-strand cDNA was synthesized to form a stable double-strand structure.

Next, we added the end repair mix to blunt the sticky ends of the double-stranded cDNA structure. This was followed by end repair, poly-A tail processing, and adaptor ligation. After the purification of cDNA, the library was enriched via PCR and quantified using TBS380 (Picogreen, Madison, WI, USA). Sequencing was performed on the Illumina Novaseq 6000 platform (Paired-end, 2 × 150 bp).

To ensure the quality of our sequencing data, we initially screened the raw data using an internal script in fastq format. We then obtained clean data using stringent filtering criteria and calculated relevant data quality indicators such as Q20, Q30, and GC content. We used only high-quality clean data for further downstream analyses.

To combine the mapping metrics of each sample, we utilized Scripture (beta2) [21] and Cufflinks (v2.1.1) [22]. We ran Scripture with default parameters and Cufflinks with ‘min-frags-per-transfrag = 0′ and ‘--library-type’ parameters, whereas other parameters were set to default. Finally, we predicted the encoding ability of novel mRNA using the Coding-Non-Coding Index (CNCI), Coding Potential Calculator (CPC), and Pfam-scan.

### 2.4. Quantification of Gene Expression Level

To quantify the expression level of genes, we used Cuffdiff (V2.1.1) to calculate the abundance of mRNA [23]. We calculated the FPKM of mRNA by summing the FPKM of the transcripts in each sample genome. The Cuffdiff program performed statistical modeling based on a negative binomial distribution to determine differential expression in digital transcripts of gene expression data. We considered statistical results to be differentially expressed when * *p* < 0.05.

### 2.5. Gene Function Enrichment Analysis

Enrichment analysis of Gene Ontology (GO), Kyoto Encyclopedia of Genes and Genomes (KEGG) pathways, and gene set enrichment analysis (GSEA) were both performed in R package clusterProfiler (V4.1.2) [24].

### 2.6. Additional Bioinformatic Methods

R package VennDiagram (V1.7.0) was employed to identify overlapping genes between different groups and generate a Venn diagram [25]. Visualization of data was performed with the R package ggplot2 (V3.3.2) [26]. Integrated bioinformatic software Unipro UGENE (V40.0) [27] was used to perform phylogenetic analysis and the online integrative tool iTOL (V5) [28] was used to visualize the phylogenetic tree. RNA secondary structure was predicted by using the RNAfold function in the Vienna RNA website [29].

### 2.7. Real-Time Quantitative PCR

In order to analyze the expression level of mRNA, a total of 0.1 μg RNA was utilized to conduct reverse transcription and produce complementary DNA (cDNA). This process was carried out with the use of the First Strand cDNA Synthesis Kit (#FSK-101, TOYOBO, Tokyo, Japan) which contained an oligo (dT) primer (5′-TTTTTTTTTTTTTTTTTTTT-3′) following the manufacturer’s guidelines. To perform quantitative real-time PCR, the SYBR Green Real-time PCR Master Mix (#11201ES03, Yeasen, Shanghai, China) and the LightCycler480 II system (Roche, Basel, Switzerland) were used. Amplification was carried out according to the following protocol: 50 °C for 2 min, 95 °C for 10 min, followed by 40 cycles of 95 °C for 15 s, 60 °C for 15 s, and 72 °C for 30 s. Using the 2^−ΔΔCt^ method, the relative expression values of the target mRNAs were normalized to those of GAPDH in each sample. All primers used in this study were listed in Table 1.

### 2.8. Western Blotting

Vero cells were infected with either PEDV strains CT-P10 or CT-P120 at an MOI of 0.1, and then lysed using cell lysis buffer (Beyotime, Shanghai, China). The lysates were subjected to boiling, following which the proteins were separated via 10% sodium dodecyl sulfate-polyacrylamide gel electrophoresis (SDS-PAGE) (Bio-Rad, Hercules, CA, USA). The resolved proteins were then transferred onto a polyvinylidene difluoride (PVDF) membrane (Millipore, Burlington, MA, USA). After blocking non-specific binding sites, the membrane was incubated overnight at 4 °C with primary antibodies against NFKB1 (Proteintech, San Diego, CA, USA, 14220-1-AP), ISG15 (Affinity, Melbourne, Australia, AB2838282), IRF3 (Proteintech, 11312-1-AP), TRAF3 (Cell Signaling Technology, Danvers, MA, USA, 4729S), PEDV-N, and GAPDH (Cell Signaling Technology) to detect respective proteins. Subsequently, the blots for NFKB1, ISG 15, IRF3, TRAF3, and GAPDH were incubated with HRP-labeled anti-rabbit IgG secondary antibody (Cell Signaling Technology), whereas the PEDV-N blot was incubated with HRP-conjugated anti-mouse IgG secondary antibody (Cell Signaling Technology). The protein bands were visualized using the ECL Plus chemiluminescent substrate (Pierce, Rockford, IL, USA).

### 2.9. Measurement of Cytokines in Cell Supernatants

Vero cells were inoculated with PEDV strains CT-P10 and CT-P120 at an MOI of 0.1. The supernatants were harvested at 6, 12, and 24 h post-infection for quantification of IL-6 (Jianglai, Shanghai, China), CCL2 (Jianglai, Shanghai, China), IL-1α (Jianglai, Shanghai, China), and IFNL1 (mlbio, Shanghai, China) using respective enzyme-linked immunosorbent assay (ELISA) kits. All measurements were performed according to the manufacturers’ instructions, and absorbance was read at 450 nm using a Model 680 Absorbance Microplate Reader (Bio-Rad, Hercules, CA, USA).

## 3. Results

### 3.1. Establishment of Cellular Model for PEDV Infection

To explore the differences in the biological properties of virulent or attenuated PEDV in vitro, Vero E6 cells were infected with either CT-P10 or CT-P120 strains both at an MOI of 0.1 for 6, 12, or 24 h, and mock-infected cells were set as a control group at each time point. Cells were subsequently collected and RNA was extracted for RNA sequencing with Illumina Novaseq 6000 platform (Figure 1A). The expression level of PEDV nucleocapsid (N) protein was confirmed through IFA, and viral titers were examined by a 50% tissue culture infectious dose (TCID_50_) assay. The results show that two PEDV strains, CT-P10 and CT-P120, both successfully infected Vero E6 cells replicated in a time-dependent manner (Figure 1B–D). However, CT-P120 has a higher rate of infection and replication capacity than PEDV CT-P10 at all time points (Figure 1C,D). The viral growth curves of the two strains in Vero E6 cells were additionally determined for viral titers at 6, 12, and 24 hpi, which showed a trend consistent with IFA results (Figure 1D). In summary, we have successfully established the Vero E6 cell model for PEDV strain CT-P10 or CT-P120 infection.

### 3.2. Assessment of RNA-Sequencing Quality

To further investigate the gene expression of PEDV CT-P10 or CT-P120 strains during infecting Vero E6 cells, transcriptomic sequencing was performed at three time points (6, 12, and 24 hpi) after infection. Each group was replicated in triplicate, resulting in a total of 27 samples. Firstly, by using the gene expression level matrix of all samples, the reproducibility and specificity of each group, were analyzed with principal component analysis (PCA). The results show that the samples within the same PEDV- or mock-infected groups are closely clustered, indicating satisfactory reproducibility within groups and evident specificity between groups (Figure 1E).

All samples were sequenced using an Illumina Novaseq 6000 platform, and a total of about 1.5 billion original reads were obtained. After filtering out the adaptor sequences and low-quality sequences, the number of clean reads of each sample ranges from 46 million to 72 million. Clean reads were then mapped to *Chlorocebus sabaeus* or PEDV reference genome, and a total of 27,985 annotated genes were obtained. Gene expression distribution of each sample was assessed by taking the logarithm of 10 of FPKM + 1, the result shows that the distribution of gene expression is basically consistent, indicating the quality of all samples is highly reliable after RNA library construction and sequencing (Appendix A). The percent of reads mapped to reference genomes ranged from 93.77% to 96.36% among all samples. No reads in mock-infected samples, CT-P10 or CT-P120-infected samples at 6 hpi were mapped to the PEDV reference genome (Appendix A). The average percent of reads mapped to PEDV reference genome in CT-P10- or CT-P120-infected groups were 17.59% or 30.06% at 12 hpi and 36.42% or 45.19% at 24 hpi (Appendix A), exhibiting a trend of rising viral reads proportion as infection progresses and higher proliferation efficiency of CT-P120 strain, which is consistent with IFA and TCID_50_ results (Figure 1B,C).

### 3.3. Weighted Gene Co-Expression Network Analysis (WGCNA)

WGCNA is primarily used to mine genes with similar expression profiles from large sample sets, identify genes in modules with high correlation with traits, and discover potential biomarkers and drug targets. By using the expression levels of 27,985 genes detected in all 27 samples, a co-expression module was constructed using the gene hierarchical cluster network tool R package, WGCNA (V1.69) [30]. The correlation matrix and adjacency matrix of PEDV (PEDV CT-P10 or PEDV CT-P120) infected samples were analyzed by WGCNA standard parameters. Six significant co-expression modules (MEblue, MEyellow, MEbrown, MEturquoise, MEgreen, and MEred) were constructed by hierarchical average chain clustering (Figure 2A and Table 2). Genes that are not classified in any other module species are classified in the MEgrey module. All samples were clustered according to their gene expression levels (Figure 2A). The correlation between the genes of each module and each group was analyzed and depicted as a heatmap (Figure 2B). The results show that the correlation coefficient between the CT-P10 infected group at 6 hpi and the MEblue is 0.73 (*p* < 1 × 10^−5^), whereas the correlation coefficient between the CT-P10 infected group at 12 hpi and the MEred group is −0.7 (*p* < 1 × 10^−5^). The heatmap in Figure 2C depicts the distances between the modules, with the MEblue and MEred being 0.27 apart (*p* < 0.01). In order to reduce noise and false correlation, and convert the adjacency matrix into a topological overlap matrix TOM (topological overlap matrix). Through hierarchical clustering of TOM matrix, the new distance matrix obtained can be used to calculate the degree of correlation between genes, and Figure 2D is obtained, in which the gene at the tip of the branch is the core gene.

### 3.4. Identification of Differentially Expressed Genes in PEDV-Infected Cells

In the process of cell infection, those genes with significant differential changes are called differentially expressed genes and may play an important role in the cell antiviral process. A total of 2963 differentially expressed genes (DEGs) were identified in the infected samples compared with the simulated infection samples with *p*-adjusted value ≤ 0.01 and | log2 (fold change) | ≥ 1 parameters. Overall, the total number of DEGs increased with the duration of viral infection in the infected group versus the mock-infected group for the two different strains. At 6 hpi, the number of differentially expressed genes in the CT-P10 strain infection group was 79, of which 26 were upregulated and 53 were downregulated, whereas no differentially expressed genes were detected in the CT-P120 strain infection group. At 12 hpi, the number of DEGs in the CT-P10 and CT-P120 strain infection groups was 479 and 13, respectively. At 24 hpi, the number of DEGs in the CT-P10 and CT-P120 strain infection groups was 1328 and 1064, respectively. The numbers of DEGs obtained in the CT-P120 strain infection group compared with the CT-P10 strain infection group at 6, 12, and 24 hpi were 127, 380, and 433, respectively (Figure 3A and Appendix A). These data indicate that infection of cultured Vero E6 cells with PEDV CT-P10 or PEDV CT-P120 causes extensive changes in host gene expression patterns.

Venn diagrams were furthermore generated to examine distinct and overlapped DEGs between subgroups infected with the same virus strain at three time points of infection or between subgroups infected with different strains but at the same time points of infection. In the group infected with strain CT-P10, 6 DEGs were present simultaneously at three time points, whereas in the group infected with strain CT-P120, 11 of the 13 DEGs at 12 hpi were also found at 24 hpi. A total of 6 DEGs were found in the three time points of the CT-P120 infected group compared to the CT-P10 infected group (Figure 3B).

### 3.5. Gene Ontology (GO) Enrichment Analysis of DEGs

The GO database is used to classify the genes in the gene set. According to the functions of the genes, GO terms could be divided into three major categories: biological process (BP), cellular component (CC), and molecular function (MF). The top 30 enriched GO terms of differentially expressed genes between CT-P10 and CT-P120 infected groups are shown in Figure 4. At 6 hpi, the most enriched GO terms of DEGs in MF include potassium ion transmembrane transport, positive regulation of cation transmembrane transport, and Rho protein signaling transduction. At 12 hpi, the most enriched GO terms in MF are DNA replication, response to the virus, DNA-templated DNA replication, defense response to the symbiont, and defense response to the virus. At 24 hpi, the most enriched GO terms in MF include response to the virus, defense response to the symbiont, defense response to the virus, cytokine-mediated signaling pathway, and negative regulation of the viral process. The top 30 enriched GO terms of DEGs between CT-P10 and mock-infected groups are shown in Appendix A–C and the top 30 enriched GO terms of DEGs between CT-P120 and mock-infected groups are shown in Appendix A–E.

### 3.6. KEGG Enrichment Analysis of DEGs

Next, we conducted a KEGG enrichment analysis on the differentially expressed genes between the two infected groups. The enriched pathways included the TNF signaling pathway, RIG-I-like receptor signaling pathway, and NOD-like receptor signaling pathway (Figure 5). In the CT-P10 infection group, the differentially expressed genes were enriched in pathways such as the TNF signaling pathway, IL-17 signaling pathway, and apoptosis at all three time points post-infection. The pathways enriched at both time points included viral protein interaction with cytokine and cytokine receptor, NF-Kappa B signaling pathway, and MAPK signaling pathway. Furthermore, the RIG-I-like receptor signaling pathway was enriched at 24 h after infection (Appendix A). However, in the CT-P120 infection group, no differentially expressed genes were found at 6 h after infection, and thus there was no pathway enrichment. At 12 h post-infection, the p53 signaling pathway and apoptosis were enriched. At 24 h post-infection, the TNF signaling pathway and MAPK signaling pathway were enriched (Appendix A).

### 3.7. GSEA Pathways Enrichment Analysis

The KEGG and GO enrichment analyses facilitated the identification of significantly altered genes and their associated biological functions and pathways. However, these traditional differential gene analysis methods focus primarily on genes that display significant changes, which might overlook genes with subtle but coordinated changes. Gene set enrichment analysis (GSEA) identifies coordinated expression trends of all genes within a pathway, not limited to the ones significantly altered. It provides us with a holistic overview of the gene expression landscape within each pathway and can thus reveal significant biological processes that may be overlooked by the traditional enrichment analysis. Hence, in this study, we employed GSEA to compare all genes between two infected groups at the same time point. Our results indicated a general downregulation of pathways in the CT-P120 infected group, particularly those related to immunity. At 6 h post-infection (hpi), the enriched pathways included proteasome, ribosome, and regulation of autophagy (Figure 6A), with the core genes of the most significantly enriched pathway in autophagy regulation (FDR = 0.2325) including ATG12, PIK3C3, GABARAPL1, ATG4C, and ATG4A (Figure 6B). At 12 hpi, enriched pathways included RIG-I-like receptor signaling pathway, NOD-like receptor signaling pathway, and toll-like receptor signaling pathway (Figure 6C), with the core genes of the most significantly enriched pathway in the toll-like receptor signaling pathway (FDR = 0.0291) being IL-1B, IL-6, MAPKs, and CXCLs (Figure 6D). At 24 hpi, the significantly enriched pathways were similar to those observed at 12 hpi, with the addition of an apoptosis pathway (Figure 6E). The core genes in the apoptosis pathway (FDR = 0.03975) included CASP3, NFKB1, MYD88, and TRAF2 (Figure 6F).

### 3.8. Immune-Related Pathways Involved in PEDV Infection

In this study, we sought to understand the role of immune-related pathways in the context of PEDV infection. To achieve this, we used GSEA to identify key pathways that are known to be fundamental to the host’s innate immune response to viral infection, including toll-like and RIG-I-like receptor signaling pathways, autophagy, apoptosis, and the cytokine-cytokine receptor interaction pathway [31,32]. These pathways were chosen due to their widespread involvement in host responses to a myriad of RNA viruses, including PEDV.

For a comprehensive view of each pathway, we selected the core genes and visualized their interactions in a pathway map (Figure 7). Following the infection with the CT-P10 virus, we noted a gradual increase in the expression levels of genes within these pathways over the course of the infection in Vero E6 cells. In contrast, infection with the CT-P120 virus resulted in significant suppression of these genes at all three time points post-infection.

This dichotomy was particularly prominent in the case of the interferon-stimulated gene response axis TRAF3-IRF3-IFNL1-ISG15 and the inflammatory factor generation axis NFKB1/MAP2K3-IL1A-CCL2. The downregulation of these axes upon CT-P120 infection suggests that the higher viral titer of CT-P120 might be attributed to its ability to suppress the expression of immune-related genes.

### 3.9. Validation of RNA-Seq Data by Quantitative Real-Time PCR

To strengthen our understanding of virus–host interaction dynamics and the host’s immune modulation during CT-P10 or CT-P120 PEDV infection, we further validated the differential expression of eight node molecules. These molecules, namely TRAF3, IRF3, IFNL1, ISG15, NFKB1, MAP2K3, IL1A, and CCL2, are notable cytokines, interferon, interferon-stimulated genes or their upstream genes closely related to viral infection. We leveraged the RNA samples from the transcriptome sequencing experiment for this qRT-PCR validation.

Upon examination, we noted a marked consistency between the expression trends observed in the transcriptome sequencing data (Figure 8A–H) and the qRT-PCR results (Figure 8I–P) for the selected genes. This congruence not only solidified the reliability of our transcriptome sequencing data but also underlined the key roles these genes play in modulating the host’s immune response during original parental or cell-attenuated PEDV infection.

### 3.10. Validation of RNA-Seq Data by Western Blotting or ELISA

We acknowledge that validation at the mRNA level alone may not sufficiently confirm the results obtained from transcriptomic studies. Therefore, to further validate the differential regulation of cell immune-related signaling pathways during infection by CT-P10 or CT-P120 PEDV strains, we carried out additional validations at the protein level using Western blotting and ELISA.

Our findings highlighted distinct differences between the infection groups of CT-P10 and CT-P120 PEDV strains, specifically in the expression levels of proteins such as TRAF3, IRF3, NFKB1, and ISG15 (Figure 9A–E). Furthermore, significant variations were observed in the levels of IFNL1, IL1A, IL6, and CCL2 interferons or other cytokines in cell supernatant through ELISA when comparing these two infection conditions (Figure 9F–I).

These findings provide further evidence suggesting that the high replication level of CT-P120 within cells may be associated with the suppression of expression of the aforementioned pathways or genes. Additionally, they underscore the reliability of our transcriptomic research, reaffirming the value of such comprehensive analytical approaches in the investigation of viral pathogenicity and host immune response dynamics.

## 4. Discussion

Since 2010, new variants of PEDV were found to spread around China and then to the world [2,33], and immunization with attenuated live vaccine has become one of the most promising and efficient means of controlling and eradicating PED. Serial subculturing viruses on cell lines is a canonical method of preparing attenuated strains. Cell-attenuated porcine epidemic diarrhea virus strains were considered promising candidates for vaccines, which could provide effective immune protection in piglets [10,34]. In the previous study, compared to the original PEDV strain CT-P10, pathogenicity experiments using CT-P120 in piglets revealed significant reductions in clinical symptoms, histopathological lesions, and intestinal PEDV antigen distribution; the piglet survival rate in the CT-P120 group was 100% [12]. Therefore, further biological studies found that the PEDV CT-P120 virus titer and virus growth curve were higher than that of PEDV CT-P10.

To further analyze the differences in their biological characteristics and explain the mechanism of viral immune evasion, transcriptomic analysis was performed on the RNA gene-level differences during Vero cell infection with CT-P10 or CT-P120 strains. Our results demonstrated that compared to the CT-P10 infection group, fewer DEGs were dysregulated in the attenuated strain CT-P120 infection group, and expression levels of immune or inflammation-related genes were wildly restrained. Our analysis, underpinned by the selection of key genes and pathways (Figure 7), provides a comprehensive view of the complex interplay between host immune response and PEDV infection. It also highlights the differential immune modulations induced by original parental or cell-attenuated PEDV, contributing to our understanding of their varying pathogenicity and virulence. This observation is consistent with findings from other studies on coronavirus infection dynamics, where broad-spectrum suppression of the innate immune response by newly emerged SARS-CoV-2 variants was observed, signifying a common strategy utilized by coronaviruses for immune evasion and viral proliferation [35,36]. Our study also contributes to the understanding of the role of ISGs in innate immunity, which is particularly relevant given that restrained ISG expression, such as ISG15 and IFITs family proteins, has been associated with coronavirus immune evasion, further emphasizing the importance of these genes as potential therapeutic targets [35,36]. Restrained gene mRNA expression of ISGs may provide evidence of coronavirus immune evasion and provide guidance for the identification of therapy targets (Figure 8). Our findings build upon our previous research, where we reported that the naturally generated virulent strain GDS01 evaded the immune system and promoted its proliferation by suppressing the TLR3/IFIT2 axis [37]. This underpins the concept of coronavirus immune evasion by imposing restrictions on immune-related gene expression. Interestingly, our results bear resemblance to another study investigating the transcriptome profiles of 2-D porcine enteroids with infection of low- or high-passage PEDV. It was found that high-passage PEDV could benefit its replication by limiting the expression of genes related to lipid metabolism [38]. Together with our findings, these studies paint a broader picture of the adaptation strategies of attenuated PEDV strains in host cells, thereby providing novel insights for the prevention and control of PED.

Significantly, our findings suggest that minor but critical mutations can dramatically influence viral characteristics such as replication capacity and stability [39,40]. Firstly, mutation of nucleotide might induce the change of viral RNA secondary structure, which could have an influence on interactions with innate host cell defenses [41]. The impact of all mutations of CT-P120 on viral RNA secondary structure flanking the mutation region (250 bp upstream and 250 bp downstream of mutation site) [42] was evaluated and compared with that of CT-P10, the results show that three mutations (A10197C, A21427C and G27710U) directly lead to significant changes in RNA secondary structure of CT-P120 strain genome (Appendix A) and the influence of other mutation was limited (data not shown). Viruses could use RNA structural motifs to avoid immune restriction [43], and this could be one of the strategies used by CT-P120 to escape immune surveillance.

A sense mutation of nucleotide could lead to a mutation in the amino acids of viral protein and influence the biological characteristics. For example, the D614G mutation in the spike protein of SARS-CoV-2 could enhance viral loads in hamsters’ upper respiratory [39], which may increase transmission risk. Compared with CT-P10, a total of 16 nucleotide mutations were detected in the CT-P120 genome, 13 of which were non-synonymous mutations, 2 in Nsp3 (A1538S, T1945M), 2 in Nsp4 (D2813G, H2925Y), 1 in Nsp6 (Y3302S), 1 in Nsp12 (V4548I), 4 in the S protein (D264A, F635R, S887R, and C1362G), 1 in each ORF3 (T45M), 1 in E (P70L), and 1 in M protein (G159D). These mutations may be the reason for the higher intracellular replication efficiency of the attenuated strain, which can be verified later through reverse genetics and other technologies, thus laying the foundation for revealing the replication and immune escape mechanism of PEDV or vaccine research and development. Among these mutation sites, H2925Y in Nsp4 was identified in vaccine strain YN144, and S887R in S protein was found in vaccine strain PT-P96, and amino acid mutation at the same site in E protein was also found as P70L in CT-P120 and P70S in vaccine strain KNU-141112 DEL5/ORF3 [4]. Moreover, the parallel amino acid mutations found in different attenuated strains demonstrate an underlying law in viral adaption to cells, which might provide instructions for future vaccine design. Interestingly, although the CT-P120 strain has become an attenuated strain for pigs, it still belongs to the G2 subtype PEDV due to its limiting variation compared with CT-P10 according to the phylogenetic analysis (Appendix A). Attenuated PEDV vaccine strain DR13 is as well as clustered into the G1 subtype along with its parental strain, variant DR13 (Appendix A).

The generation of attenuated virus is the basis for the preparation of live attenuated vaccine [19]. The practical relevance of our study extends to the development of novel PEDV therapeutics. Given the observed mutations in CT-P120, a better understanding of these changes could potentially pave the way for the design of more effective and targeted vaccine strains. Additionally, reverse genetic technology could be employed to quickly validate the function of these mutation sites in immune escape, providing vital data for the rapid preparation of attenuated virus strains. Therefore, our research not only elucidates the intricate mechanisms of viral immune evasion but also provides insights for future therapeutic advancements.

## 5. Conclusions

In conclusion, our study provides a comprehensive insight into the host cell response to CT-P120, a highly cell-passaged porcine epidemic diarrhea virus (PEDV) strain, in contrast with its parental strain, CT-P10. We observe CT-P120′s attenuated virulence and strategic employment of immune evasion during replication. This is primarily achieved by suppressing multiple interferon-associated and interferon-stimulated gene (ISG)-related pathways, facilitating its proliferation. The findings significantly advance our understanding of PEDV’s adaptive mechanisms and escape strategies, with the downregulation of crucial immune-related genes lending CT-P120 higher viral titers. These insights lay the groundwork for the development of more effective PEDV vaccines and affirm the potential of cell-passaged strains such as CT-P120 in vaccine design and infectious disease control.

## 6. Limitations

Although our study unveils different aspects of immune-related pathway regulation during original parental or cell-attenuated PEDV infection, we recognize some limitations that necessitate further exploration. The distinct immunomodulatory effects of CT-P10 and CT-P120 have not been investigated using gene overexpression or silencing techniques. Similarly, the biological significance of mutation sites in cell-attenuated PEDV strains awaits verification through reverse genetics. These limitations, however, provide a compelling direction for our future research endeavors. A deeper understanding of these aspects is not only crucial to unraveling the intricacies of PEDV pathogenicity and immune evasion but also holds promise in expediting the development of effective vaccines against PEDV.

## Figures and Tables

**Figure 1 viruses-15-01426-f001:**
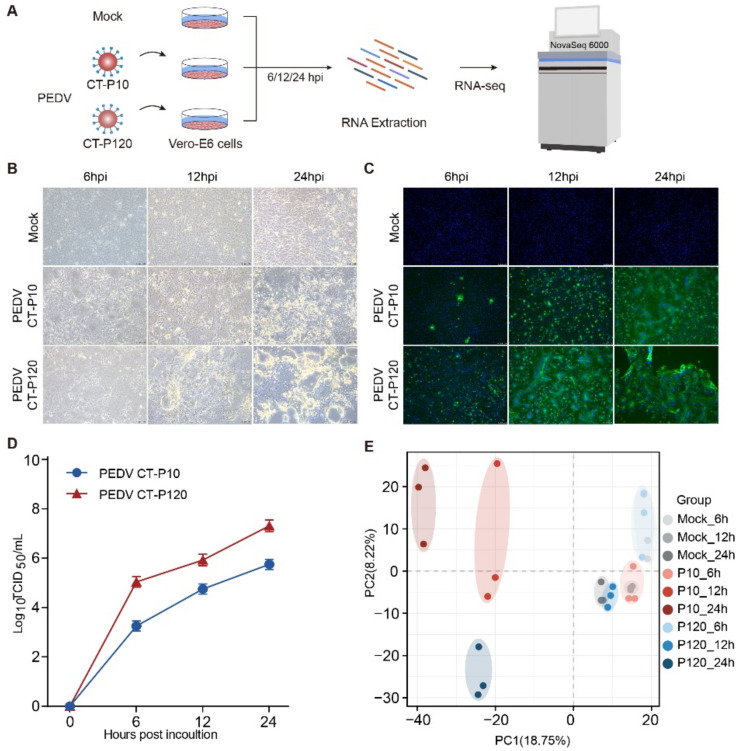
Basic information of this study. (**A**) Diagram of RNA-seq experimental design. (**B**) Cells were observed under a bright field through a microscope. (**C**) Identification of PEDV infection by indirect immunofluorescence. The nucleus was stained by 4′,6-diamidino-2-phenylindole (DAPI), blue; the PEDV N protein was stained by fluorescein isothiocyanate (FITC), green. (**D**) PEDV proliferation curve determined using TCID_50_. (**E**) Principal component analysis of each sample.

**Figure 2 viruses-15-01426-f002:**
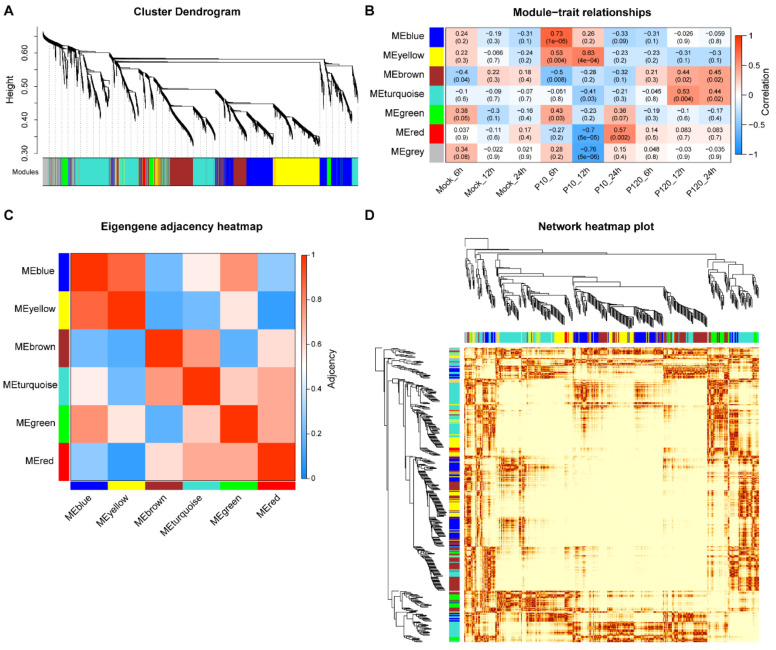
Weighted correlation network analysis (WGCNA). (**A**) Genes are divided into modules according to the expression trend of genes, where branches represent a gene, a color represents a module, and if the color is gray, it represents a gene that is not assigned to a specific module. (**B**) Show the correlation size between modules and modules, each column or row represents a module, the color in the figure represents the correlation size between modules, red represents the greater correlation between modules, and green represents the small correlation between modules. On the left or above is a tree view of module clustering, and the closer the two module branches are, the more related the two modules are. (**C**) The *x*-axis represents different groupings, the *y*-axis represents different modules, the number on the left side of the figure indicates the number of genes in the module, and each set of data on the right represents the correlation coefficient and significance of the module with the phenotype and the significance *p*-value (in parentheses), red represents the positive correlation between the module and the phenotype, and blue represents the negative correlation between the module and the phenotype. (**D**) Module eigengene adjacency heatmap. Module eigengenes (ME) in this analysis are defined as the first principal component of a coexpression module matrix. The heatmap shows the relatedness of the 13 co-expression modules (ME1-ME13) identified by WGCNA with red being highly related and blue being not related.

**Figure 3 viruses-15-01426-f003:**
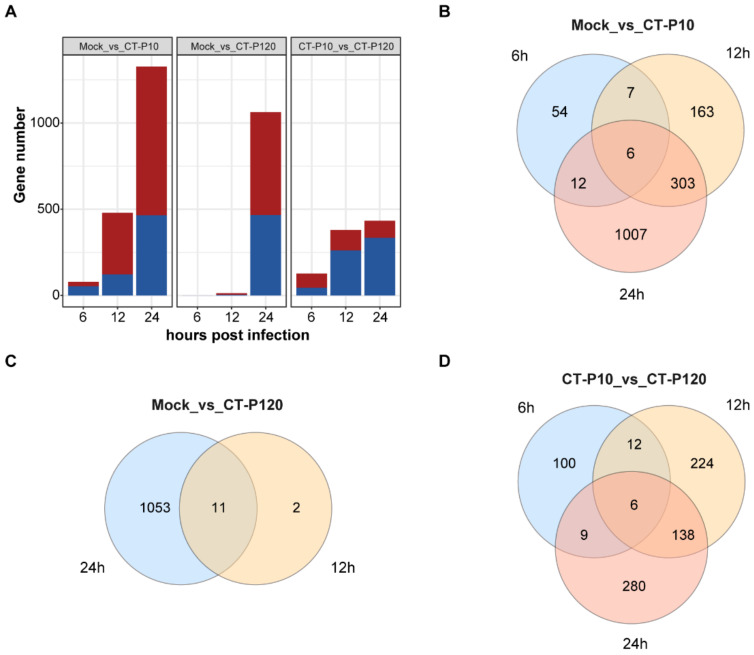
Analysis of differentially expressed genes. (**A**) Histograms of differentially expressed genes number. Bars in blue indicated downregulated gene numbers, and bars in red indicated upregulated genes number. (**B**) Venn diagrams of differentially expressed genes between mock and CT-P10 infection groups at three time points. (**C**) Venn diagrams of differentially expressed genes between mock and CT-P120 infection groups at two time points. (**D**) Venn diagrams of differentially expressed genes between CT-P10 and CT-P120 infection groups at three time points.

**Figure 4 viruses-15-01426-f004:**
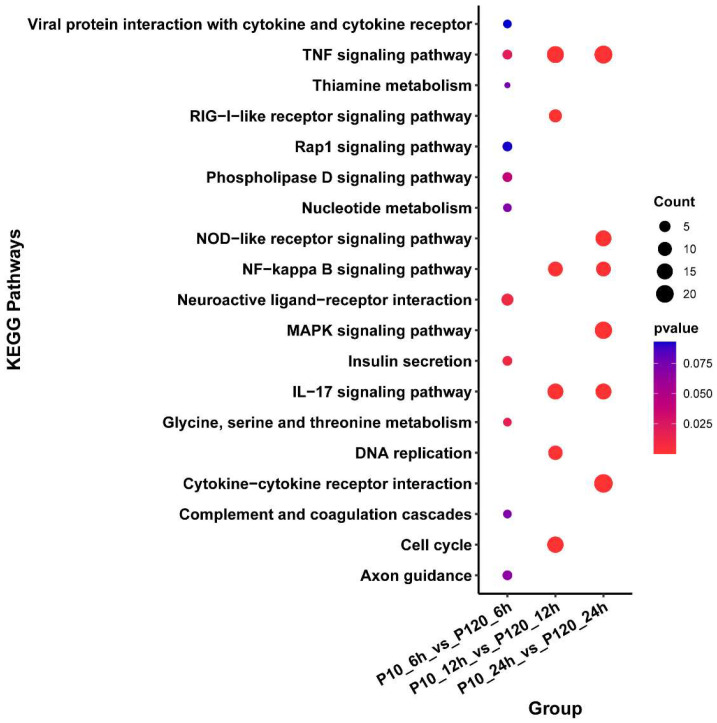
Bubble plot of KEGG pathways in CT-P120 infected groups compared with CT-P10 infected groups. The size of each point represents the number of DEGs, and the color of each point represents the *p*-value of enriched KEGG pathways.

**Figure 5 viruses-15-01426-f005:**
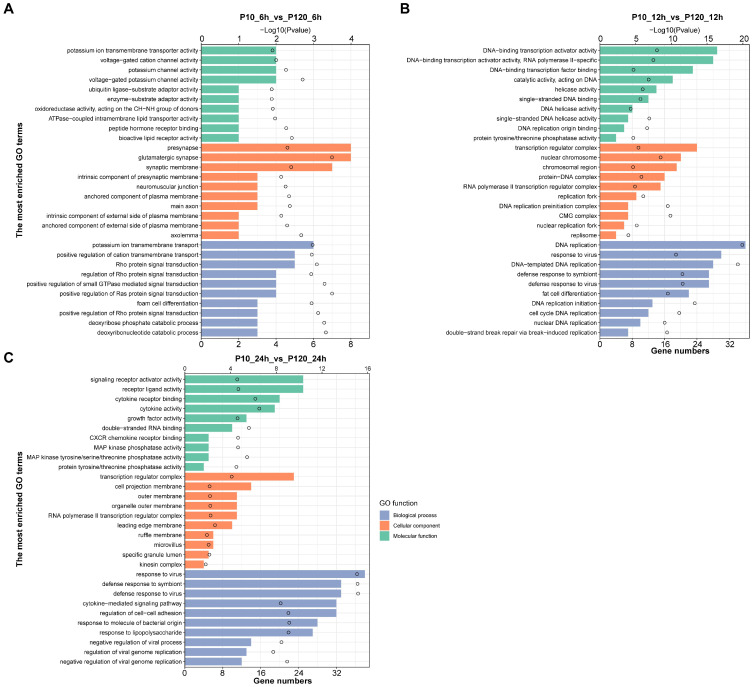
Bar plot of GO terms in CT-P120 infected groups compared with CT-P10 infected groups. Bars in green indicated biological process terms, bars in orange indicated cellular component terms, and bars in blue indicated molecular function terms. *x*-axis at bottom indicated gene numbers in each GO term and data were shown as bars, *x*-axis at top indicated *p*-value of GO terms and data were shown as circles, and *y*-axis indicated different GO terms. (**A**) Bar plot of GO terms in CT-P120 infected group versus CT-P10 infected group at 6 hpi. (**B**) Bar plot of GO terms in CT-P120 infected group versus CT-P10 infected group at 12 hpi. (**C**) Bar plot of GO terms in CT-P120 infected group versus CT-P10 infected group at 24 hpi.

**Figure 6 viruses-15-01426-f006:**
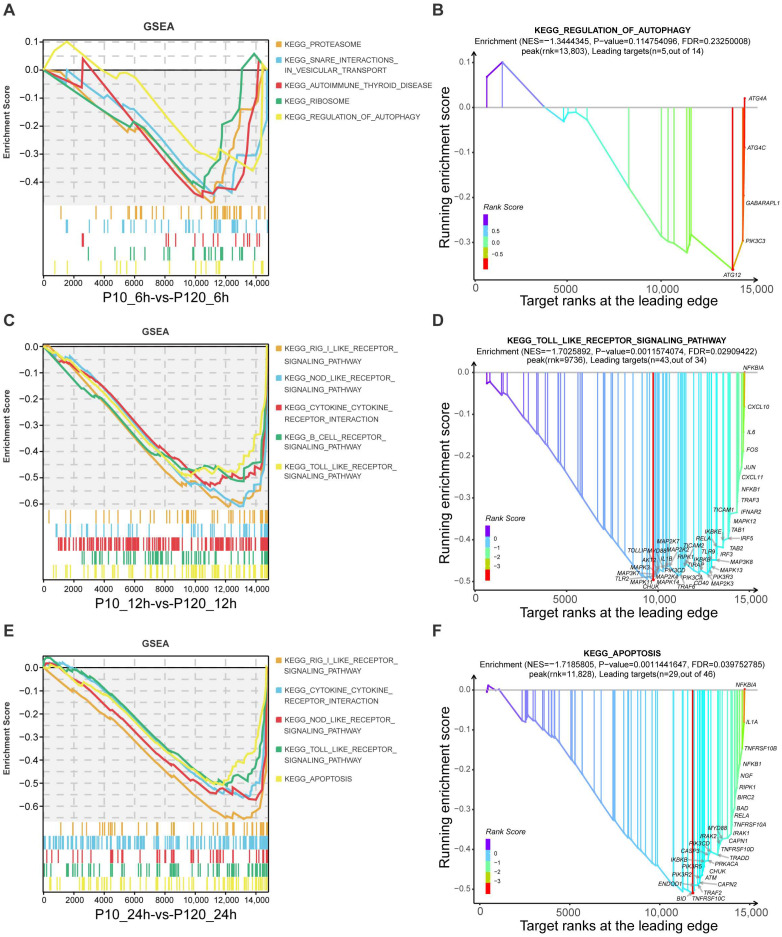
Top 5 enriched viral infection- or immune-related KEGG pathways in GSEA results. (**A**,**C**,**E**), top 5 enriched viral infection- or immune-related KEGG pathways in GSEA results of CT-P120 infected cells versus CT-P10 infected cells at 6, 12, 24 hpi. (**B**,**D**,**F**), top 1 enriched viral infection- or immune-related KEGG pathways with gene annotations in GSEA results of CT-P120 infected cells versus CT-P10 infected cells at 6, 12, 24 hpi.

**Figure 7 viruses-15-01426-f007:**
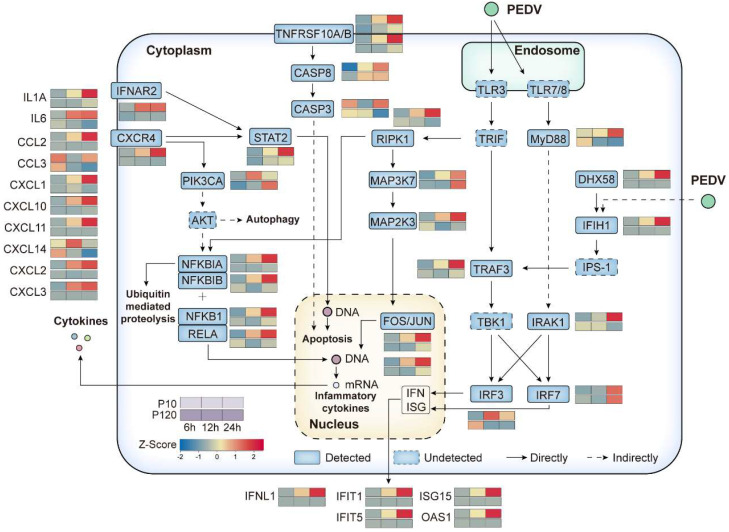
Map of genes in immune-related pathways with heatmaps involved in PEDV infection. Rounded rectangles with solid lines indicate detected genes in transcriptomic data, and rounded rectangles with dotted lines indicated undetected genes in transcriptomic data. Heatmaps next to the rounded rectangles indicate the gene expression levels in a z-score form, the upper lanes indicate the CT-P10 infection group, and the lower lanes indicate the CT-P120 infection group. Solid lines with arrows indicate the direct interaction, and dotted lines with arrows indicate the indirect interaction.

**Figure 8 viruses-15-01426-f008:**
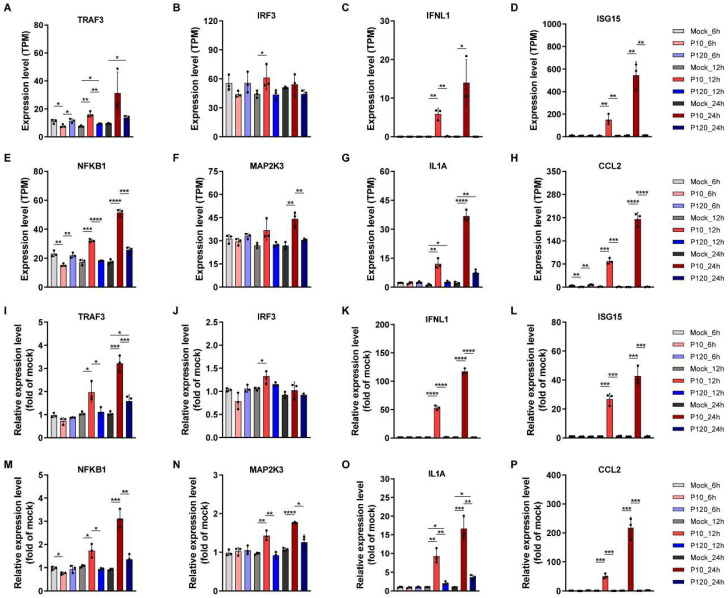
Validation of RNA-sequencing data by quantitative real-time PCR. Cultured Vero E6 cells were mock-infected or infected with PEDV strain CT-P10 or CT-P120 at MOI of 0.1. At 6, 12, and 24 hpi, samples were harvested and relative mRNA expression levels of indicated genes were measured by quantitative real-time PCR. (**A**–**H**) Expression level of eight selected genes in transcriptomic data. (**I**–**P**) Expression level of eight selected genes examined by RT-qPCR. Data are expressed as mean ± SEM from three independent experiments. Data were analyzed using the Mann–Whitney test. *: *p* < 0.05, **: *p* < 0.01, ***: *p* < 0.001, ****: *p* < 0.0001. SEM, standard error of mean.

**Figure 9 viruses-15-01426-f009:**
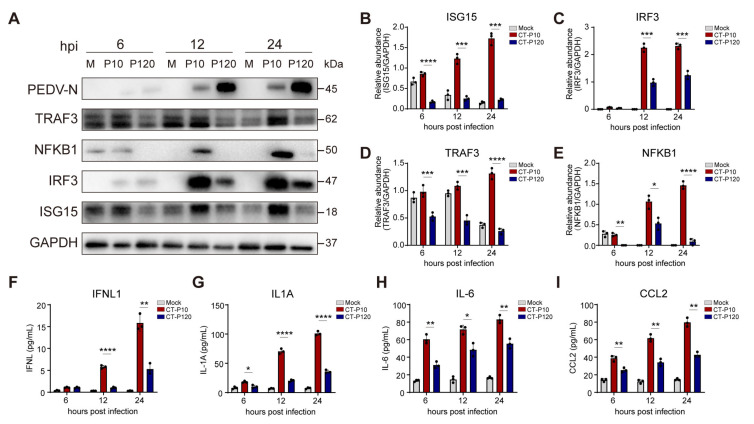
Validation of RNA-sequencing data by Western blotting and enzyme-linked immunosorbent assay. Cultured Vero E6 cells were either mock-infected or infected with PEDV strain CT-P10 or CT-P120 at an MOI of 0.1. The cells were harvested at 6, 12, and 24 hpi for protein analysis. (**A**) Western blotting analysis of protein samples extracted from the treated cells, M: marker, P10: CT-P10, P120: CT-P120. (**B**–**E**) Relative protein expression levels of ISG15, IRF3, TRAF3, and NFKB1, respectively, as compared to GAPDH. (**F**–**I**) Quantitative analysis of IFNL1, IL-1α, IL-6, and CCL2 concentrations in the cell supernatant as measured by ELISA. Data are expressed as mean ± SEM from three independent experiments. The data were analyzed using the Mann–Whitney test. Statistical significance is indicated as follows: *: *p* < 0.05, ***: *p* < 0.01, ***: *p* < 0.001, ****: *p* < 0.0001. SEM, standard error of mean.

**Table 1 viruses-15-01426-t001:** Sequences of all primers used in this study.

Gene	Primer	Sequence	Product Length (bp)
TRAF3	Forward	5′-CCTTGTTCCGATTTGGAGGTG-3′	300
	Reverse	5′-TGACCCGGCTCCATTCTGTG-3′	
IRF3	Forward	5′-TTGTGACCTCAGGAGTTGGG-3′	249
	Reverse	5′-GCTTCAGTGGGTTTTCACGG-3′	
IFNL1	Forward	5′-CGGGAATTGGGACCTAAGGC-3′	274
	Reverse	5′-GCCAGGGGACTCCTTTTCGG-3′	
ISG15	Forward	5′-CACGGCCATGGGTAGGGA-3′	266
	Reverse	5′-TCCTCACCAGGATGCTCAGT-3′	
NFKB1	Forward	5′-GGCTACCCTGGCACAGAAAT-3′	291
	Reverse	5′-TCATCCCGGAGCTCGTCTAT-3′	
MAP2K3	Forward	5′-CCATCGGAGACAGGAACTTTGA-3′	260
	Reverse	5′-GACGTCCAAGTCCATGAGCA-3′	
IL1A	Forward	5′-GCCCGCAATCAAAGCATCAT-3′	217
	Reverse	5′-GTGTCTCAGGCAGCTCCTTC-3′	
CCL2	Forward	5′-GTGTCCTAAAGAAGCAGTGATCTTC-3′	198
	Reverse	5′-TCTGAGGGTATTTAGGGCAAGT-3′	
GAPDH	Forward	5′-CGGAGTGAACGGATTTGGC-3′	248
	Reverse	5′-CACCCCATTTGATGTTGGCG-3′	

**Table 2 viruses-15-01426-t002:** Numbers of genes clustered in six WGCNA modules.

Module	Blue	Yellow	Brown	Turquoise	Green	Red
Gene number	1154	923	926	1470	238	48

## Data Availability

The raw data of RNA-sequencing have been uploaded to the National Center for Biotechnology Information (NCBI) (https://www.ncbi.nlm.nih.gov (accessed on 2 April 2023)) and the accession numbers are from SRR23279294 to SRR23279320.

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
