# Peer review of "Transcriptome Profiling of Vero E6 Cells during Original Parental or Cell-Attenuated Porcine Epidemic Diarrhea Virus Infection"

_viruses, 2023, doi:10.3390/v15071426_

Round 1

Reviewer 1 Report

Peng et al. here reported a comparative transcriptomic profiling of Vero E6 cells infected with parental strain (CT-P10) and the cell-attenuated strain (CT-P120) of porcine epidemic diarrhea virus at the indicated times after infection and found that CT-P10 induced numerous innate immune-related pathways whereas CT-P120 attenuated some innate immune signals, such as the Toll/RIG-I-like receptor signaling and the IL-17 signaling pathways. The experimental design is well-done, and the result data is convinced. Some minor editing of English language is required before accepted for publication in Viruses.

Some grammatical errors in the manuscript need to be modified. 

Reviewer 2 Report

In this manuscript, Peng et al performed a comparative transcriptomic analysis of Vero E6 cells infected with either the original parental strain (CT-P10) or the cell-attenuated strain (CT-P120) of PEDV (porcine epidemic diarrhea virus) at different time points. The authors revealed that CT-P10 activated several innate immune-related pathways whereas CT-P120 attenuated their activation. Overall, it could be a valuable contribution to the field. However, several points require attention and should be addressed as described below.

1. The authors focued on the different response of the several innate immune-related pathways at mRNA level upon CT-P10 or CT-P120 infection. But the authors should confirm the roles of these pathways using gene depletion or overexpression. Otherwise, that would greatly undercut the importance of this study.

2. Illumina HiSeq 2500 sequencing system and Illumina Novaseq 6000 platform was mentioned in line 120 and line 170, respectively. Please clarify.

3. Panel labelings in Figure 1 are incorrectly marked. There are two 1C.

4. In section 2.3, the method for libary preparation and sequencing is still unclear. Please add more details.

Minor editing of English language required

Reviewer 3 Report

In this study, the authors investigated the transcriptomic features of Vero E6 cells during infection with either parental or cell-adapted attenuated strains of porcine epidemic diarrhea virus (PEDV) through detailed experiments and data analysis. Overall, the design and execution of the study are solid, and the results hold scientific value. However, the following concerns should be addressed in revision. 

1. In the results section, the authors used multiple approaches (e.g., GO enrichment, KEGG enrichment, and GSEA) to analyze differentially expressed genes. However, the relationship and complementary nature of these methods have not been adequately discussed. I suggest that the authors engage in more discussion in this regard to help readers better understand the role of these methods in the study. 

2. Section 3.8 of the results section focuses on immune-related pathways involved in the PEDV infection process. The authors need to provide more detailed explanations of the roles of these pathways during infection and their changes at different time points and under different viral infection conditions. Furthermore, discussing how to utilize this information to understand the pathogenicity and virulence of the virus is crucial. 

3. In section 3.9 of the results, the authors validated some differentially expressed genes using qRT-PCR. However, the authors need to provide more detailed explanations as to why they chose these specific genes for validation and their roles in the viral infection and immune response processes. 

4. In the discussion section, the authors should discuss their findings in the context of other related studies and provide a more comprehensive comparison of the results. This would help to strengthen their conclusions and highlight the importance of their study within the field. 

5. The authors should elaborate on the potential implications of their findings for the development of new PEDV vaccines or therapeutic strategies. This would help to underline the practical relevance of their research. 

6. The discussion could benefit from a more in-depth examination of the limitations of the study, as well as possible future research directions that could help address these limitations or expand upon the authors' findings.

Minor editing of English language is required in revision.

Round 2

Reviewer 2 Report

The review has no further concerns.